# Non-Enzymatic Glucose Biosensor Based on Highly Pure TiO_2_ Nanoparticles

**DOI:** 10.3390/bios11050149

**Published:** 2021-05-11

**Authors:** Hongin Jeong, Jhongryul Yoo, Seokyung Park, Jiling Lu, Sungho Park, Jeseung Lee

**Affiliations:** 1Department of Life Science and Chemistry, Daejin University, 1007 Hoguk Road, Pocheon-si 11159, Korea; wjdghddlsdl1@naver.com (H.J.); whdfuf1345@naver.com (J.Y.); 2Department of Chemistry and Research Institute of Basic Sciences, Kyung Hee University, 26 Kyungheedae-ro, Dongdaemun-gu, Seoul 02447, Korea; tjrud1015@naver.com (S.P.); lujiling@naver.com (J.L.)

**Keywords:** titanium dioxide, nanoparticle, deposition, glucose, biosensor

## Abstract

This study proposes a non-enzymatic glucose sensor fabricated by synthesizing high-purity TiO_2_ nanoparticles in thermal plasma and depositing it directly on a substrate and then depositing chitosan–polypyrrole (CS-PPy) conductive polymer films by electrochemical method. The structural properties of the deposited TiO_2_ nanoparticles were analyzed by X-ray diffraction (XRD) and dynamic light scattering (DLS) system. The chemical composition and structural properties of the TiO_2_ nanoparticle layer and the conductive polymer films were confirmed by X-ray photoelectron spectroscopy (XPS) spectra and scanning electron microscope (SEM). The glucose detection characteristics of the fabricated biosensor were determined by cyclic voltammetry (CV). CS-PPy/TiO_2_ biosensor showed high sensitivity of 302.0 µA mM^−1^ cm^−2^ (*R*^2^ = 0.9957) and low detection limit of 6.7 μM. The easily manufactured CS-PPy/TiO_2_ biosensor showed excellent selectivity and reactivity.

## 1. Introduction

Glucose is an essential metabolite of all living organisms. Diabetes is a metabolic disorder caused by impaired insulin secretion or dysfunction, characterized by the elevated blood sugar levels that must be maintained at a constant level by insulin and glucagon produced by the pancreas. Chronic hyperglycemia caused by diabetes leads to damage and dysfunction of each organ of the body, especially microvascular complications in the retina, kidney, and nerves, as well as giant vascular complications such as arteriosclerosis and cardiovascular and cerebrovascular diseases, resulting in increased mortality [1,2,3]. Although the incidence of complications due to the onset and progression of diabetes can be reduced by weight loss or medication, the most important strategy is thorough blood sugar management and prevention. The normal range of glucose levels in the blood is 4.4–6.6 mM (80–120 mg/100 mL), and higher or lower levels of glucose in the blood indicate a problem with insulin secretion or the glucose levels not being controlled even though insulin is normally secreted [4].

Since the first electrochemical glucose sensor was designed by Clark and Lyons in 1962, more advanced biosensors for detecting glucose have been developed by many researchers [5]. Updike and Hicks developed a method for measuring the glucose concentration in biological fluids by immobilizing the glucose oxidase (GOx) on the electrode surface [6]. When glucose and protein enzymes combine, hydrogen peroxide (H_2_O_2_) is produced. The concentration of glucose in sample is measured by detecting the movement of electrons generated by the decomposition of hydrogen peroxide on the electrode surface. In early glucose biosensors, the low selectivity to glucose due to the interference of other substances present in the sample to the enzymes was a serious problem to overcome [4,7,8]. Nowadays, glucose dehydrogenase (GDH) or GOx-based electrochemical biosensors are considered as one of the most efficient glucose detecting devices with the advantages of high selectivity, sensitivity, and low detection limit. Although enzyme-based biosensors have excellent advantages in terms of convenience and performance as a regular blood glucose measurement device, there is a need to be complemented in terms of thermal and chemical stability because the enzymes can be affected by pH of blood samples, temperature, the partial pressure of oxygen, and humidity during the measurement and long-term storage [1,9,10].

To overcome the drawbacks of enzyme-based biosensors, many researchers are conducting ongoing work on non-enzymatic biosensors using nanomaterials. Enzyme-based sensors are difficult to mass produce and hold in long-term storage due to disadvantages such as the complexity of enzyme immobilization process and the instability of biologically active substances. Most non-enzyme sensors, however, are relatively easy for mass production due to the advantages including relatively simple structure, manufacturing process, and easy quality control [3]. Non-enzymatic glucose sensors have been developed using various metals, metal oxides, alloys, metal complexes, and carbon materials as the electrochemical catalysts for the oxidation of glucose in blood samples [1,4,11,12,13,14,15,16,17,18,19,20]. Various metal oxide NPs including copper oxide (Cu_2_O), titanium oxide (TiO_2_), and iron oxide (Fe_3_O_4_) in their original or modified form have been used for glucose oxidation [21,22,23,24,25,26,27,28]. Among the metal oxide NPs in particular especially, the interest for titanium dioxide (TiO_2_) NP is being amplified due to its unique properties such as high surface area and high catalytic efficiency, which can improve the interaction between biomolecules and electrode surfaces [29,30,31,32,33]. Recently, a glucose sensor using a nanocomposite film of polypyrrole–chitosan–TiO_2_ with a detection limit of 614 µM has been reported [34].

Herein, the high purity TiO_2_ nanoparticles that were deposited on the fluorine-doped tin oxide coated glass slide (FTO) using thermal plasma deposition based on the glucose biosensor were fabricated. Deposited TiO_2_ nanoparticles were electrochemically deposited by chitosan–polypyrrole (CS-PPy) conductive polymer to prepare the CS-PPy/TiO_2_ sensor. Enhanced sensitivity and selectivity of glucose biosensors was obtained by using high purity and small particle sizes of thermal plasma deposited TiO_2_ nanoparticles.

## 2. Materials and Methods

### 2.1. Materials and Chemicals

TiCl_4_ (ReagentPlus^®^, 99.9%), chitosan (CS), D-(+)-glucose, sodium *p*-toluenesulfonate (Na *p*-TS), pyrrole (Py), and fluorine-doped tin oxide coated glass slide (FTO) were purchased from Sigma-Aldrich Co. (Yongin, Korea). Ar and O_2_ gases (99.999%) were purchased from Sinyang Co. (Seoul, Korea).

### 2.2. Characterization Methods

Field emission scanning electron microscope (FE-SEM, Hitachi, Model SU 8220, Tokyo, Japan) and high-resolution transmission electron microscope (HR-TEM, Tecnai G2-F20, FEI Company, Hillsboro, OR, USA) equipped with EDS PV9761 detector at the acceleration voltage of 200 kV were used to investigate the morphology and structural properties of the deposited nanocomposite films. The NPs in the films were analyzed by X-ray photoelectron spectroscopy (XPS) measurements with a PHI 5800 ESCA System (Physical Electronics) equipped with a hemispherical energy analyzer and the monochromatized Al*Kα* X-ray source of 250 W. A potentiostat instrument (PARSTAT 2263, Princeton Applied Research, Oak Ridge, TN, USA) was used for the electrochemical measurements. The FTO glass electrode, platinum mesh, and Ag/AgCl were used as a working electrode, counter electrode, and reference electrode, respectively.

### 2.3. Fabrication and Deposition of TiO_2_ Nanoparticles

To fabricate and deposit the crystalline TiO_2_ NPs, we used a self-developed form of equipment capable of depositing precursors (TiCl_4_) using a thermal plasma directly on a fluorine-doped tin oxide coated glass slide (FTO) substrate [35]. Ar carrier gas was injected into a bubble-type canister containing TiCl_4_ to inject a Ti precursor into a stabilized plasma flame. The resulting TiO_2_ NP was injected into the chamber and deposited on a FTO substrate.

### 2.4. Deposition of CS-PPy Film on the Deposited TiO_2_ NPs on FTO

Electrochemical deposition of CS-PPy conductive film on TiO_2_ NP layer was proceeded by the modified reported method [29]. To prepare the reaction solution, we dissolved 0.5 g of CS in 50 mL of acetic acid (1 M) under continuous stirring at room temperature for 1 h, followed by ultrasonic treatment for 2 h. A total of 10 mL of Na *p*-TS (2 wt %) and a certain amount of pyrrole were added to the previous solution as dopants with stirring for 10 min. As shown in Figure 1, FTO with immobilized TiO_2_ NPs was added to the prepared reaction solution. The CS-PPy conductive polymer film was deposited on the TiO_2_ layer using the three-electrode cell system at a potential range of −1 to +1.2 V (vs. Ag/AgCl) with a scan rate of 50 mV s^−1^. The CS-PPy/TiO_2_ nanocomposite films were washed repeatedly with deionized water and dried at the ambient temperature.

## 3. Results

Crystalline TiO_2_ NPs with high purity were deposited on FTO substrate using a self-developed form of equipment that can deposit precursors (TiCl_4_) using a thermal plasma directly on a substrate. The injected Ti precursor (TiCl_4_) was quickly mixed with the O_2_ radical and the swirling gas to produce TiO_2_ NPs. The produced NPs were injected into the chamber and deposited on the surface of FTO substrate [35].

Figure 2a shows the TEM images of the deposited TiO_2_ NPs with the size of prepared NPs. The average size of TiO_2_ NPs deposited through a microwave plasma process was measured at around 20 nm. It can be confirmed that the highly pure TiO_2_ NPs were synthesized by the result of TEM-EDX observation, as shown in Figure 2b. The variation in size of the NPs according to the plasma processing time was also investigated. Dynamic light scattering (DLS) results showed that the average particle sizes were about 20 nm, even as the processing time increased (Table 1).

Figure 3 shows the XRD patterns of TiO_2_ nanoparticles deposited on FTO through microwave plasma process. As shown in Figure 3a, the weight fraction of deposited crystalline TiO_2_ NPs was 66% for the anatase phase and 34% for the rutile phase. As shown in Figure 3b, even through varying the deposition time to 4 or 16 min, we found that the crystallinity of TiO_2_ was reproduced and the ratio of phases did not change. In addition, it can be seen that there was no change in crystallinity even after heat treatment at 450 °C for 24 h to immobilize the deposited TiO_2_ NP layer on to FTO.

Figure 4a,b are the cross-section images of the TiO_2_ NP layer deposited on the FTO and the chitosan–polypyrrole (CS-PPy) polymer film deposited on the TiO_2_ NP layer by electrochemical method, respectively. As shown in Figure 4a, the TiO_2_ NP layer was deposited on the FTO substrate with a thickness of about 2 μm. Figure 4b shows that the CS-PPy conductive polymer film was deposited on the surface of the TiO_2_ NP layer through an electrochemical method. Figure 4c exhibits a schematic diagram showing that the TiO_2_ NP layer was deposited on the FTO substrate and the CS-PPy film was deposited thereon. During the electrochemical deposition, polypyrrole was connected to chitosan through hydrogen bonding, and CS-PPy/TiO_2_ nanocomposite conductive polymer film was formed through hydrogen bonding with TiO_2_ NPs and Ti–nitrogen ligand bonding. The chemical structure of the formed CS-PPy/TiO_2_ film is shown in Figure 4d.

Figure 5 shows the X-ray photoelectron spectroscopy (XPS) analysis results for C, O, N, and Ti, the main components of the CS-PPy/TiO_2_ nanocomposite conductive film. Figure 5a is the depth profile of the film. It can be seen that as the etching time increased, the CS-PPy layer on the surface was reduced by shaving, and the TiO_2_ NP layer was exposed, according to the increased Ti and O fractions and the decreased C and N fractions. Figure 5b,c show the XPS spectrum before and after etching for 900 s and the change in atomic concentration ratio for each element. After etching for 900 s, the fraction of TiO_2_ NP increased more than the CS-PPy layer and the fraction of Ti and O became closer to the ratio of Ti/O = 1:2.

For the detection of glucose using the CS-PPy/TiO_2_ nanocomposite conductive film as a sensor, we used a three-electrode cell system the same as the electrochemical deposition of CS-PPy layer with a scan rate of 50 mV s^−1^ in the range of −0.8 to +1.0 V (vs. Ag/AgCl). Glucose detection was measured by cyclic voltammetry under basic conditions (0.1 M NaOH at pH = 8.1). As shown in Figure 6a, the peak observed at about +0.13 V was according to the oxidation of glucose. Figure 6b shows a schematic diagram for the oxidation reaction of glucose on the sensor surface during the glucose detection. Glucose is oxidized to gluconolactone on the surface of sensor electrode [1], and the generated electrons can be measured by changing the current to determine the concentration of glucose in solution.

Figure 7a shows the current change according to the glucose concentration in aqueous solution using the CS-PPy/TiO_2_ electrode as a sensor using the chronoamperometry method, and the inset is a calibration curve for the current change according to the varied concentration of glucose. The glucose concentrations in the aqueous solutions were 1, 3, 5, 7, 9, and 11 mM, and the current was measured by applying the potential of +0.13 V for each solution. The measured currents were proportional to the concentration of glucose in each aqueous solution. The sensitivity and the limit of detection (LOD) of the CS-PPy/TiO_2_ sensor were determined as 302 μA mM^−1^ cm^−2^ and 6.7 μM, respectively. The detection limit was calculated using the following Equation (1):LOD = 3 × sb/m(1)
where sb is the standard deviation of the blank and m is the sensitivity between the concentration and current (S/N = 3, confidence level = 98.1%). Figure 7b is a chronoamperometry graph comparing the sensitivity of the CS-PPy/TiO_2_ sensor for glucose and some substances such as ascorbic acid, uric acid, and cholesterol present in blood sample. The potential of +0.13 V was applied, and the concentrations of each substances were 0.1 mM. The large current change was detected for glucose, but the similar low current changes were observed for all comparative materials, implying the high selectivity of CS-PPy/TiO_2_ sensor for glucose.

The amperometric responses of CS-PPy/TiO_2_ nanocomposites on FTO were also measured at an applied potential of +0.13 V (vs. Ag/AgCl) with consecutively increasing the concentration of glucose. As shown in Figure 8, the amperometric response of CS-PPy/TiO_2_ electrode for glucose increased linearly by increasing the concentration of glucose.

In Figure 9, the sensitivities for the detection of glucose of the CS-PPy/TiO_2_ glucose sensor produced by the plasma process and were compared by the sol–gel method. Using the cyclic voltammetry method, we measured the basic aqueous solutions of 1 mM glucose at a scan rate of 50 mV s^−1^ in the potential range of −0.8 to +1.0 V (vs. Ag/AgCl).

The sensitivities and detection limits of sensors fabricated using various metal oxides were compared. As shown in Table 2, the sensitivity of the TiO_2_ sensors was not very high, but it was satisfactory. In particular, the TiO_2_ sensor has the advantage that blood sample dilution is not required because the linear range of the TiO_2_ sensor covers the entire range of glucose concentrations that are actually present in human blood. Importantly, the glucose sensor fabricated by the plasma process had higher sensitivity and lower detection limit than that fabricated by the sol–gel method under the same condition. This result seems to be because the TiO_2_ NPs deposited by the plasma process have higher purity and crystallinity than the TiO_2_ NPs produced by the sol–gel method and have a larger surface area due to the relatively smaller particle size.

## 4. Conclusions

A highly sensitive, non-enzymatic glucose sensor based on the high-purity TiO_2_ nanoparticles was successfully fabricated. TiO_2_ nanoparticles were synthesized in thermal plasma and deposited directly on an FTO substrate followed by deposited the chitosan-polypyrrole (CS-PPy) conductive polymer films by electrochemical method to obtain the nanocomposites. The deposited TiO_2_ NPs were highly pure and had 66% of anatase phase. The average particle sizes of TiO_2_ NPs were about 20 nm and the thickness of TiO_2_ layer was about 2 μm. The chemical composition and structural properties of the TiO_2_ NP layer and the conductive polymer films were confirmed by X-ray photoelectron spectroscopy (XPS) spectra and scanning electron microscope (SEM). CS-PPy/TiO_2_ glucose biosensor showed high sensitivity of 302.0 µA mM^−1^ cm^−2^ (*R*^2^ = 0.9957) and low detection limit of 6.7 μM in basic solution. The higher sensitivity and lower detection limit of the glucose biosensor prepared by deposition using the plasma process rather than by sol–gel method may have been due to the higher purity and larger surface area of the TiO_2_ NPs deposited by the plasma process in comparison with those by the sol–gel method.

## Figures and Tables

**Figure 1 biosensors-11-00149-f001:**
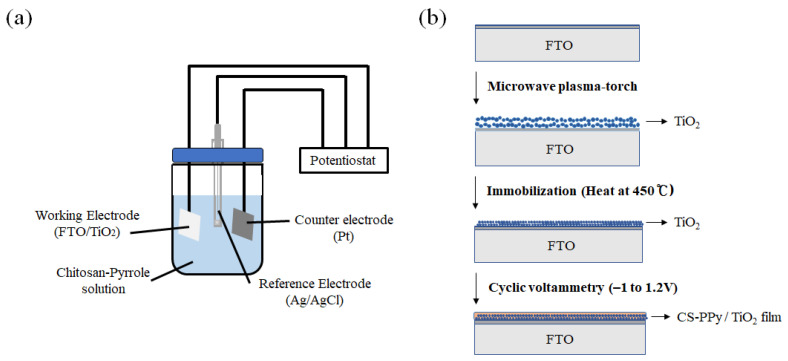
Schematic diagram of (**a**) a three-electrode measuring system and (**b**) the preparing CS-PPy/TiO_2_ nanocomposite films on FTO.

**Figure 2 biosensors-11-00149-f002:**
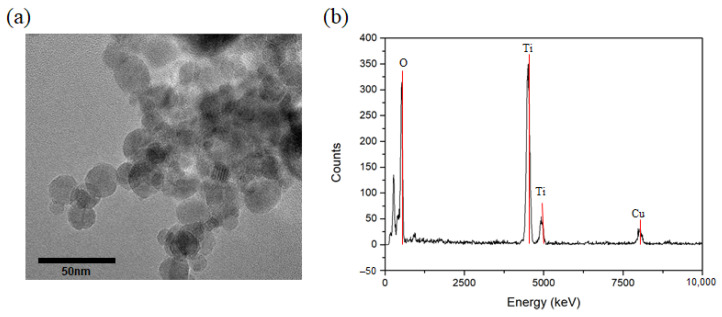
(**a**) TEM image and (**b**) EDX data of TiO_2_ nanoparticles prepared by thermal plasma process.

**Figure 3 biosensors-11-00149-f003:**
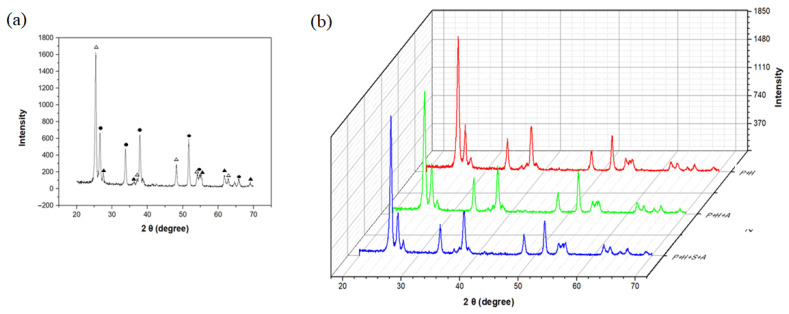
XRD patterns of (**a**) deposited TiO_2_ NPs on FTO electrode (Δ: anatase phase, ▲: rutile phase, ●: FTO) and (**b**) after deposition and after immobilization (in order from top to bottom, after 4 min of deposition, after 16 min of deposition, and after heat treatment (450 °C)).

**Figure 4 biosensors-11-00149-f004:**
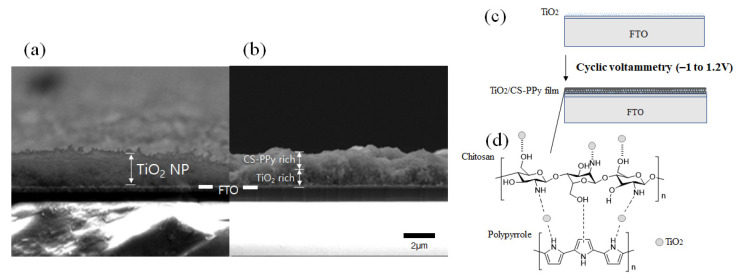
SEM images of the cross-section of (**a**) TiO_2_ deposited on the FTO electrode and (**b**) CS-PPy conductive polymer film deposited on (**a**). (**c**) Schematic procedure of depositing CS-PPy conductive polymer on TiO_2_ deposited on FTO and (**d**) chemical structure of a CS-PPy/TiO_2_ films.

**Figure 5 biosensors-11-00149-f005:**
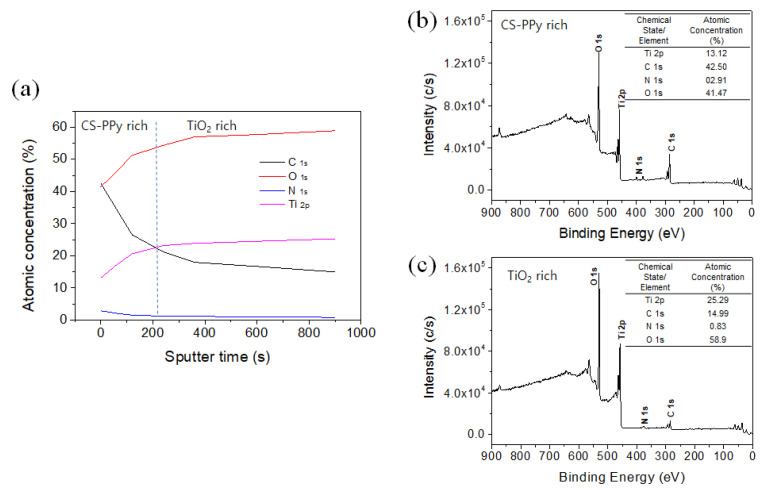
(**a**) XPS depth profile of CS-PPy/TiO_2_ glucose sensor. XPS survey spectra and atomic concentration fraction of (**b**) before sputtering and (**c**) after 900 s sputtering.

**Figure 6 biosensors-11-00149-f006:**
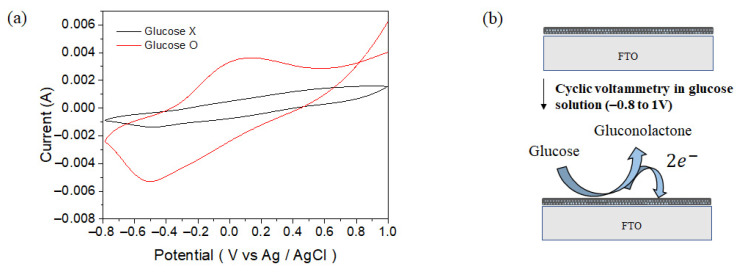
Cyclic voltammograms of CS-PPy/TiO_2_ sensor in 0.1 M NaOH solution (**a**) with 1 mM glucose and without glucose and (**b**) reaction on the sensor surface during the glucose detection.

**Figure 7 biosensors-11-00149-f007:**
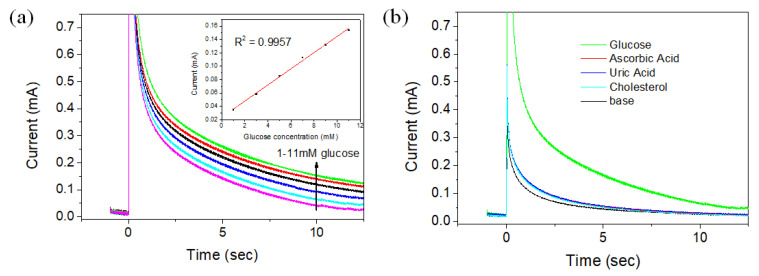
Chronoamperograms of CS-PPy/TiO_2_ glucose sensor in 0.1 M NaOH solution at +0.13 V with (**a**) varied glucose concentration (the inset shows the steady-state calibration curve) and (**b**) the different substances.

**Figure 8 biosensors-11-00149-f008:**
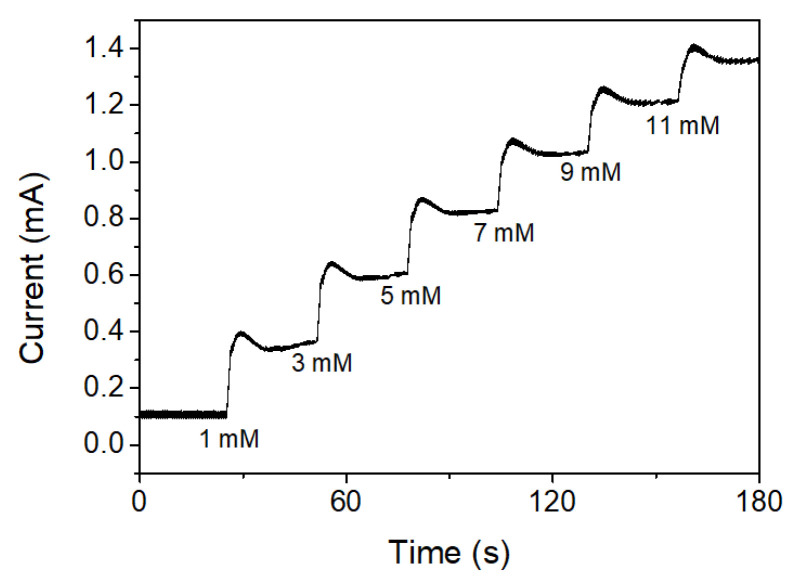
Amperometric responses to the consecutive addition of glucose solution in 0.1 M NaOH solution at +0.13 V.

**Figure 9 biosensors-11-00149-f009:**
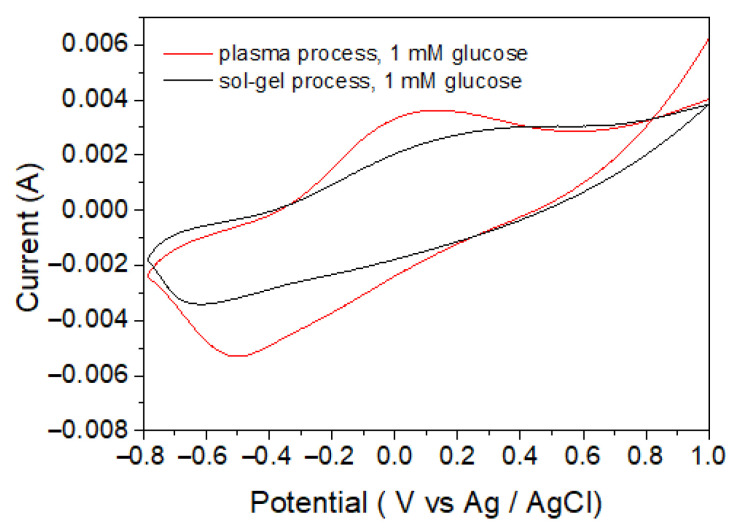
Cyclic voltammograms of CS-PPy/TiO_2_ prepared by plasma process and sol–gel process.

**Table 1 biosensors-11-00149-t001:** Dependence on the deposition time for the average particle sizes of deposited TiO_2_ NPs.

Deposition Time (min)	Average Particle Size (nm)
2	19.2
4	18.7
8	20.9
16	19.7

**Table 2 biosensors-11-00149-t002:** Comparison of biosensor characteristics according to the metal oxide NPs.

Metal Oxide	Sensitivity (µA mM^−1^ cm^−2^)	Linear Range (mM)	LOD (µM, S/N = 3)	Reference
NiO	43.9	0.001–0.11	0.16	[22]
Cu_2_O	62.29	0.05–6.75	37	[23]
Cu_2_O	185	0.01–6	0.05	[24]
Co_3_O_4_	520.7	0.005–0.8	0.13	[25]
Fe_3_O_4_	238.7	0.5–7	15	[26]
Fe_2_O_3_	1382.8	0.0025–0.58	0.58	[27]
SnO_2_	1930	0.050–0.500	13.35	[28]
TiO_2_ ^a^	0.008	1–14	614	[29]
TiO_2_ ^b^	254.4	1–9	62	this work
TiO_2_ ^c^	302.0	1–11	6.7	this work

^a^ Electrochemical deposition. ^b^ Sol–gel method. ^c^ Plasma thermal deposition.

## Data Availability

Not applicable.

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
