# Peer review of "Non-Enzymatic Glucose Biosensor Based on Highly Pure TiO2 Nanoparticles"

_biosensors, 2021, doi:10.3390/bios11050149_

Round 1
Reviewer 1 Report
The present paper fabricated the electrochemical enzymeless glucose biosensor composed of TiO2 nanoparticle. They provided the useful data for glucose biosensor construction. The manuscript may be eventually publishable, but still it requires revision as indicated.
Also, there are some questions raised by reviewer
- In Figure 1b, it is hard to recognize the schematic diagram, please draw it again more clearly.
- In Figure 2a, the TEM image didn’t show the uniformity of synthesized nanoparticle. Please show it again with large scale.
- The comparison table provides the useful information for readers. Please suggest it with other group works.
- For confirming the clinical applications, please carry out clinical samples for the next submission.
Author Response
Responses to the Reviewer 1
Manuscript ID: biosensors-1186242
Title: " Non-enzymatic glucose biosensor based on highly pure thermal plasma deposited TiO2 nanoparticles"
Thank you for the valuable and constructive comments and suggestions that are of great help in preparing the revised manuscript with improved quality. The changes in manuscript were highlighted using the “Track Changes” function. Please see the attachment.
Comments and Suggestions for Authors #1
The present paper fabricated the electrochemical enzymeless glucose biosensor composed of TiO2 nanoparticle. They provided the useful data for glucose biosensor construction. The manuscript may be eventually publishable, but still it requires revision as indicated.
Also, there are some questions raised by reviewer
- In Figure 1b, it is hard to recognize the schematic diagram, please draw it again more clearly.
[Response]
As reviewer recommended, we redraw the Figure 1b to be seen more clearly (line 118).
- In Figure 2a, the TEM image didn’t show the uniformity of synthesized nanoparticle. Please show it again with large scale.
[Response]
As reviewer recommended, we replace the TEM image with large scale in Figure 2 (line 137).
- The comparison table provides the useful information for readers. Please suggest it with other group works.
[Response]
Thanks for your valuable comment and constructive recommendation. The non-enzymatic glucose sensor characteristics of various nanostructured metal oxides reported by other groups were added to Table 2 and compared through line from 243 to 254. The sensitivity of the TiO2 sensors is not very high, but it is satisfactory. In particular, the TiO2 sensor has the advantage that blood sample dilution is not required because the linear range of the TiO2 sensor covers the entire range of glucose concentrations which are actually present in human blood. Importantly, the glucose sensor fabricated by the plasma process had higher sensitivity and lower detection limit than that fabricated by the sol-gel method under the same condition. This result seems to be because the TiO2 NPs deposited by the plasma process have higher purity and crystallinity than the TiO2 NPs produced by the sol-gel method, and have the larger surface area due to the relatively smaller particle size.
- For confirming the clinical applications, please carry out clinical samples for the next submission.
[Response]
Thanks for your valuable comment and constructive recommendation. The electrical characteristics of the glucose sensor developed through this work show sufficient characteristics for commercialization. In addition, the process is much simpler than the conventional complex process of fabricating nanostructures, and compatibility with semiconductor processes is high. The glucose sensor related to this work is currently being developed using 8” substrate for practical use, so it is difficult to disclose relevant clinical information at this time. Clinical data can be released after product development.
Reviewer 2 Report
The authors developed the TiO2 NP-based non-enzymatic glucose sensor. They characterized the sensor performance quantitatively and demonstrated sensitivity and selectivity. However, I cannot understand the advances compared to previous article cited as Ref.29 which is also submitted to MDPI. Therefore, I think that this article is not acceptable to Biosensors.
[Comments]
Line 75
What is the role of CS-PPy layer?
Line 178
Did the authors optimize pH value of the measurement solution? Why pH 8.1 was selected for this study?
Line 201
How do the authors think about selectivity of the present sensor against other saccharides in the human blood?
Line 239
Do the size of TiO2 NPs and its layer thickness affect sensor performance? Did the authors optimize these parameters?
Author Response
Responses to the Reviewer 2
Manuscript ID: biosensors-1186242
Title: " Non-enzymatic glucose biosensor based on highly pure thermal plasma deposited TiO2 nanoparticles"
Thank you for the valuable and constructive comments and suggestions that are of great help in preparing the revised manuscript with improved quality. The changes in manuscript were highlighted using the “Track Changes” function. Please see the attachment.
Comments and Suggestions for Authors #2
The authors developed the TiO2 NP-based non-enzymatic glucose sensor. They characterized the sensor performance quantitatively and demonstrated sensitivity and selectivity. However, I cannot understand the advances compared to previous article cited as Ref.29 which is also submitted to MDPI. Therefore, I think that this article is not acceptable to Biosensors.
[Response]
Conventional nanostructured metal oxide manufacturing methods, such as the sol-gel method or electrochemical deposition, have disadvantage that require additional heat treatment to remove these chemical substances because they use a solution in which various chemical substances are mixed. The important goal of this study is to fabricate and evaluate a non-enzymatic glucose sensor by depositing high-purity nanoparticles directly onto a substrate using our proprietary nanoparticle thin film deposition technology.
The non-enzymatic glucose sensor characteristics of various nanostructured metal oxides reported by other groups were added to Table 2 and compared through line from 243 to 254. In Ref 29, the glucose biosensor was fabricated by electrochemical deposited TiO2 nanoparticles. The biosensor's sensitivity and limit of detection were reported as 0.008 µA·mM−1 cm−2 and 614 µM, respectively. In this study, the sensitivity and limit of detection were much improved by direct deposition of TiO2 nanoparticles on the substrate using plasma thermal deposition technique.
This result is thought to be due to the fact that the nanoparticles produced through electrochemical deposition used in the reference have lower crystallinity than the deposition using plasma used in this study, and the tail of the band gap is deep, causing electron trapping. It is also believed that the reason is that the amount of nanoparticles is relatively insufficient due to the deposition of conductive polymers. Nanoparticles manufactured through the proprietary method developed in this laboratory are high purity and crystalline nanoparticles that exhibit good semiconductor properties. In addition, it is estimated that this is because the nanoparticles are deposited directly on the sensor, which is relatively less contaminated and the density of nanoparticles is higher than that of electrochemical deposition. As a result of this difference, the fabricated sensor showed higher sensitivity than the sensors manufactured using sol-gel method or electrochemical deposition by eliminating the tail of the band gap of metal oxide nanoparticles.
[Comments]
Line 75
What is the role of CS-PPy layer?
[Response]
The role of the CS-PPy layer is a conductive polymer that transfers electrons received by TiO2 nanoparticles to the analysis electrode at high speed. Without the conductive polymer, the electrons obtained by the nanoparticles cannot be quickly transferred to the analysis electrode, resulting in a rapid decrease in the current value.
Line 178
Did the authors optimize pH value of the measurement solution? Why pH 8.1 was selected for this study?
[Response]
Thanks for your valuable comment. The reason we evaluated the sensor's performance at pH 8.1 is that we aimed to commercialize the sensor in the field, taking into account that the pH of normal human blood is about 7.4.
Line 201
How do the authors think about selectivity of the present sensor against other saccharides in the human blood?
[Response]
Thanks for your valuable comment and constructive recommendation. In blood stream, other saccharides including fructose and lactose are present. The analysis of other sugars is thought to be a necessary study depending on the purpose and use. In this study, only glucose, which has a direct effect on insulin, was evaluated, and no other saccharides were analyzed. The electrical characteristics of the glucose sensor developed through this work show sufficient characteristics for commercialization. In addition, the process is much simpler than the conventional complex process of fabricating nanostructures, and compatibility with semiconductor processes is high. The glucose sensor related to this work is currently being developed using 8” substrate for practical use. During the development, we will do the experiments for the selectivity to the other saccharides.
Line 239
Do the size of TiO2 NPs and its layer thickness affect sensor performance? Did the authors optimize these parameters?
[Response]
Thanks for your valuable comment. The high-purity crystalline nanoparticle deposition technology applied as the core process of this study is not yet capable of controlling the effective particle size as shown in Table 1, and research on improvement is currently being carried out. Although the average particle size was changed at the 10-50 nm level by increasing the deposition time up to 1 h, significant change in properties was not observed. Moreover, when the thickness of the nanoparticle layer was varied by controlling the deposition time, the amount of current slightly increased as the thickness of the nanoparticle layer increased, but the differences was ignorable. Those results may due to the limitation of exposing surface of TiO2 nanoparticles to the solution by the conductive polymer surrounding the nanoparticle layer.
Round 2
Reviewer 1 Report
Authors answered the all questions well. No reivision needed.
Reviewer 2 Report
I think that this revised article is acceptable to "Biosensors" because the authors sincerely replied to my questions.